# OpenReview forum: "$Q\sharp$: Provably Optimal Distributional RL for LLM Post-Training"
_NeurIPS.cc/2025/Conference — NeurIPS 2025 poster_

### Official Review · Reviewer_H6aA · 2025-06-29

**Clarity:** 2
**Significance:** 2
**Originality:** 3
**Rating:** 4
**Confidence:** 2

**Summary:**

This manuscript proposes a novel value-based RL algorithm of Q# for LLM post-training. Concretely, Q# employes an offline way to collect data and update the value model, which yields more accurate value estimation and thus better performance. Theoretically imporvement are provided to show the rationale of Q#. Emprical study is conducted on Llama and Qwen verifies that Q# is superior to ref model and CD.

**Questions:**

This paper focuses on KL regularization in LLM training. However, many recent RL-based post-training algorithms for LLMs have removed the KL term, as it has been observed to hinder the learning of CoT reasoning [1]. The manuscript would be more convincing if the authors could empirically demonstrate the benefit of KL regularization in enhancing reasoning capabilities, particularly in long-form generation.

&nbsp;


[1] *Yu, Qiying, et al. "Dapo: An open-source llm reinforcement learning system at scale." arXiv preprint arXiv:2503.14476 (2025).*

**Ethical Concerns:**

["NO or VERY MINOR ethics concerns only"]

**Final Justification:**

The authors provided more empirical results and clarify the experimental settings.

**Paper Formatting Concerns:**

There is no formatting concerns.

**Quality:**

2

**Strengths And Weaknesses:**

## Strengths

1. The paper introduces a novel value-based RL algorithm, Q\#, designed to enhance LLM post-training.

2. Theoretical analysis is provided to support the effectiveness of Q\# and justify its design.

3. The proposed method is evaluated on multiple LLM backbones, including LLaMA and Qwen, demonstrating its general applicability and superiority.


## Weaknesses

1. The paper is difficult to follow in its current form. It is unclear which baseline algorithm Q\# builds upon, or how Q\# is derived step by step. For example, the formulation in Eq.~(5) closely resembles that of DPO, yet the connection is not well explained.

2. The experimental comparison is limited and lacks strong baselines. Q\# is only compared against the reference model (pre-trained but not post-trained) and CD. It would significantly strengthen the paper to include comparisons with more widely-used RL methods such as PPO or GRPO.

3. The empirical evaluation is restricted to two relatively simple math reasoning datasets (MATH and GSM8K). Additional evaluations on more challenging benchmarks such as AIME would provide stronger evidence for reasoning.

---

> ### Author Rebuttal · Authors · 2025-07-30
>
> Dear Reviewer H6aA,
>
> Thank you so much for your review and constructive suggestions for us to improve the paper. We respond to individual points below.
>
> > The paper is difficult to follow in its current form. It is unclear which baseline algorithm Q# builds upon, or how Q# is derived step by step. For example, the formulation in Eq.~(5) closely resembles that of DPO, yet the connection is not well explained.
>
> **Authors' reply:** Thank you for this valuable feedback. We agree that better clarifying the derivation of Q♯ and its relationship to DPO will strengthen the paper. Below, we provide a clearer derivation and explain the connection and key differences.
>
> **Derivation of Q♯:** Q♯ is derived from the standard KL-regularized RL objective in deterministic MDPs, as given in Eq. (1). Following classical results [1], the optimal policy at step $h$ takes the form (Eq. (2)): $\pi^{\star,\eta}\_h(y|x) \propto \pi^{\text{ref}}\_h(y|x) \exp(\frac{1}{\eta} Q^{\star,\eta}\_h(x, y))$ where $Q^{\star, \eta}\_h(x, y)$ is the soft action-value function. For deterministic MDPs, such as LLM decoding, we can recursively expand $Q^{\star, \eta}\_h(x, y)$ using the deterministic transition function (concatenating token $y$ to the prefix $x$). This yields Eq. (4) and substituting back the optimal policy expression yields Eq. (5), which defines a token-level policy that reweights $\pi^{\text{ref}}$ by the expected exponentiated cumulative reward-to-go.
>
> **Connection to DPO [2] (and the H = 1 case):** While the form of Eq. (5) resembles DPO’s policy expression, the derivation and scope are fundamentally different. DPO starts with the same KL-regularized RL objective but, without exploiting the deterministic transition structure, it operates at the **sequence level**, corresponding to the **one-step case (H = 1)**. Its policy is
> $\pi_r(y|x) = \frac{1}{Z(x)} \pi^{\text{ref}}(y|x) \exp(\frac{1}{\beta} r(x,y))$ where $y$ is a full completion and $Z(x)$ is the partition function over all possible sequences [2]. When $H = 1$, our $Q^{\star}$ simply equals the reward $r$, and the DPO formula follows as a special case. Additionally, DPO partition function is intractable to normalize, typically requiring pairwise preference data to bypass it (Bradley-Terry modeling).
>
> In contrast, Q♯ explicitly solves the **multi-step KL-regularized RL problem**. By leveraging deterministic MDP structure, we derive a tractable, closed-form expression for the optimal *token-level* policy in Eq. (5). Empirically, as shown in star-graph experiments (Appendix Figure 5), the difference between these paradigms matters in practice. DPO, being policy-based and sequence-level, cannot fix the shortcut behavior, while value-based methods like Q♯ achieve near-perfect accuracy by explicitly modeling the reward-to-go and guiding token-level decisions.
>
> In summary, Q♯ is not built upon DPO or any other baseline. It is an independent derivation from first principles of KL-regularized RL in deterministic MDPs [1,3-5]. Compared to DPO, Q♯ addresses the more general and challenging $H > 1$ case, and to the best of our knowledge, it is the first algorithm for LLM post-training that provably guides the reference policy using the optimal KL-regularized Q-function in a tractable token-level form. We will revise the paper to make these distinctions and derivation clearer.
>
> > The experimental comparison is limited and lacks strong baselines. Q# is only compared against the reference model (pre-trained but not post-trained) and CD. It would significantly strengthen the paper to include comparisons with more widely-used RL methods such as PPO or GRPO.
>
> **Authors' reply:** Thank you for the helpful comment! Following your suggestion, we conduct additional experiments on the MATH dataset using PPO as a baseline.
>
> In these experiments, Q♯ is instantiated with a Qwen 2.5 1.5B model and used to guide both: (1) the original Qwen 2.5 7B reference model, and (2) the same model after PPO post-training. As shown in the table below, Q♯ consistently improves performance in both settings. Notably, when applied to the PPO-trained policy, Q♯ further increases accuracies and reduces the KL divergence — indicating improved alignment with $\pi^{\text{ref}}$ and better performance. The results suggest that Q♯ can also be complementary to policy-based approaches.
>
> We additionally note a qualitative distinction: PPO improves pass@1 but slightly reduces maj1@8, indicating that its generations tend to be lower entropy and less diverse. Q♯, in contrast, improves both metrics.
>
> |Dataset|MATH||||
> |-|:-:|:-:|:-:|:-:|
> |Methods|$\pi^{\text{ref}}$|$\pi^{\text{ref}}$ + $Q\sharp$|PPO|PPO + $Q\sharp$|
> |pass@1 ↑|58.6|61.9|68.4|**71.1**|
> |maj1@8 ↑|72.8|**74.8**|72.4|73.4|
> |KL-Divergence ↓|-|**6.46**|69.52|60.53|
>
> In terms of efficiency, Q♯ is significantly lighter to train. PPO requires approximately 20 hours on 4×H100 GPUs, whereas Q♯ training completes in roughly 5 hours on a single H100 GPU, thanks to its supervised learning objective and the use of a much smaller model. These findings suggest that Q♯ can effectively enhance performance, demonstrating its practical advantage as a complementary lightweight module.
>
> > The empirical evaluation is restricted to two relatively simple math reasoning datasets (MATH and GSM8K). Additional evaluations on more challenging benchmarks such as AIME would provide stronger evidence for reasoning.
>
> **Authors' reply:** Thank you for the suggestion. We conduct additional evaluation on AIME-24 using the same Q♯ model checkpoint (Qwen 2.5 1.5B) trained on MATH to guide the Qwen 2.5 7B reference model, without any further finetuning. The results are shown below. We observe that Q♯ significantly improves performance over the reference policy, despite being trained on a different dataset, indicating generalization capability. Furthermore, Q♯ consistently outperforms the CD baseline in both accuracy metrics while maintaining a lower KL divergence, supporting its effectiveness.
>
> |Dataset||AIME-24||
> |-|:-:|:-:|:-:|
> |Methods|$\pi^{\text{ref}}$|CD|$Q\sharp$|
> |pass@1 ↑|9.3|13.5|**14.1**|
> |maj1@8 ↑|16.7|16.7|**20.0**|
> |KL-Divergence ↓|-|9.95|**9.23**|
>
> In addition, following Reviewer 2Dby’s suggestion on studying other potential applications of Q♯ beyond math reasoning, we conducted experiments on QuALITY [6], a multiple-choice QA benchmark with context passages sourced from Project Gutenberg. We experimented with two settings: Qwen 2.5 1B guiding Qwen 2.5 7B and Llama 3.2 1B guiding Llama 3.1 8B. The results suggest that Q♯ can also consistently improve performance on QA tasks for both Qwen and Llama.
>
> |$\pi^{\text{ref}}$||Qwen 2.5 7B|||Llama 3.1 8B||
> |-|:-:|:-:|:-:|:-:|:-:|:-:|
> |Methods|$\pi^{\text{ref}}$|CD|$Q\sharp$|$\pi^{\text{ref}}$|CD|$Q\sharp$|
> |pass@1 ↑|64.5|64.2|**68.1**|73.5|75.1|**75.9**|
> |maj1@8 ↑|72.0|66.3|**73.3**|79.3|79.3|**81.1**|
> |KL-Divergence ↓|-|12.32|**7.90**|-|9.23|**8.88**|
>
> **KL regularization may hinder CoT reasoning**
>
> **Authors' reply:** Thank you for raising this and highlighting concurrent work such as DAPO [7], which we will discuss and cite in the final version. We agree that the role of KL-regularized RL in post-training—particularly for long-form CoT reasoning—remains nuanced, and that in some settings, strong constraints on the reference policy could limit the exploratory behavior needed for emergent reasoning capabilities.
>
> Nevertheless, KL-regularized RL problem remains a valuable and widely used formulation that underpins many practical algorithms, including DPO [2], CD [8], and VAS [9]. Our goal in this paper is to revisit this fundamental setting and address it in a principled and provably optimal way. Prior algorithms such as CD and VAS can be viewed as approximate or sub-optimal methods for KL-regularized RL, while Q♯ directly solves for the optimal solution in a tractable, token-level form.
>
> From a theoretical perspective, KL regularization plays a crucial role in tractability. In particular, it avoids the need for Bellman completeness assumption, which is typically required for convergence in standard (unregularized) RL [10]. Our analysis shows that under KL regularization, one can obtain provable convergence under the much milder realizability condition. Thus, even when the function class is far from Bellman complete, KL regularization offers favorable theoretical properties. We will clarify our scope and motivation in the final version, and thank the reviewer again for highlighting this distinction.
>
> **Citations**
>
> [1] Ziebart, Brian D., et al. "Maximum entropy inverse reinforcement learning." Aaai. Vol. 8. 2008.
>
> [2] Rafailov, Rafael, et al. "Direct preference optimization: Your language model is secretly a reward model." Advances in neural information processing systems 36 (2023): 53728-53741.
>
> [3] Domingo-Enrich, Carles, et al. "Adjoint matching: Fine-tuning flow and diffusion generative models with memoryless stochastic optimal control." arXiv preprint arXiv:2409.08861 (2024).
>
> [4] Piché, Alexandre, et al. "Probabilistic planning with sequential monte carlo methods." International Conference on Learning Representations. 2018.
>
> [5] Li, Xiner, et al. "Derivative-free guidance in continuous and discrete diffusion models with soft value-based decoding." arXiv preprint arXiv:2408.08252 (2024).
>
> [6] Pang, Richard Yuanzhe, et al. "QuALITY: Question answering with long input texts, yes!." arXiv preprint arXiv:2112.08608 (2021).
>
> [7] Yu, Qiying, et al. "Dapo: An open-source llm reinforcement learning system at scale." arXiv preprint arXiv:2503.14476 (2025).
>
> [8] Mudgal, Sidharth, et al. "Controlled decoding from language models." arXiv preprint arXiv:2310.17022 (2023).
>
> [9] Han, Seungwook, et al. "Value augmented sampling for language model alignment and personalization." arXiv preprint arXiv:2405.06639 (2024).
>
> [10] Munos, Rémi, and Csaba Szepesvári. "Finite-Time Bounds for Fitted Value Iteration." Journal of Machine Learning Research 9.5 (2008).

---

> > ### Comment · Reviewer_H6aA · 2025-08-03
> >
> > Thank you for the response, which addresses some of my concerns. However, I remain unconvinced regarding the experimental results.
> >
> > 1. Regarding the first table in the response: although Q# appears significantly more efficient than PPO, PPO still achieves higher scores than Q#. While PPO+Q# surpasses PPO alone, this improvement seems unrelated to efficiency.
> >
> > 2. Regarding the second table on AIME24: I would like to see additional baselines (e.g., PPO or GRPO) and evaluations on larger LLMs (e.g., 7B models). Moreover, how were the reported scores obtained? Do they reflect the best scores from a single run/training, or an average across runs?

---

> > > ### Author Response · Authors · 2025-08-03
> > > **Response to Reviewer H6aA**
> > >
> > > Dear Reviewer H6aA,
> > >
> > > Thank you for reading our rebuttal and we are glad to hear it has addressed some of your concerns. We are more than happy to provide additional clarification.
> > >
> > > **Q1.**
> > >
> > > The reviewer is correct that PPO achieves higher pass@1 than Q♯, albeit at the cost of lower maj@8 and greater divergence from $\pi^{\text{ref}}$. The drop in maj@8 suggests reduced response diversity, which has been observed in prior work as collapsing the output distribution [1,2]. While increased determinism can be beneficial for certain applications, it may also lead to reduced generalization, making PPO outputs less ideal as sources for generating new data or for further training.
> > >
> > > Regarding efficiency, we emphasize that the **same** Q♯ checkpoint—trained using rollouts from $\pi^{\text{ref}}$ only—is reused to guide both $\pi^{\text{ref}}$ and PPO models. This means Q♯ can be trained independently of PPO, without requiring access to PPO-generated rollouts. The improvement seen in PPO+Q♯ thus stems entirely from this lightweight and reusable Q♯ module, which introduces minimal additional training cost.
> > >
> > > In this sense, we view Q♯ not as a competing alternative to PPO, but as a practical and efficient complement: it improves performance over PPO with negligible overhead, while preserving closer alignment with the reference policy.
> > >
> > > **Q2.**
> > >
> > > Thank you for your suggestion and we have conducted additional evaluation on AIME-24 using PPO baseline, and the updated results are provided below.
> > >
> > > We would also like to clarify that the previously reported AIME-24 results were already based on **7B models**. Specifically, \$\pi^{\text{ref}}\$ refers to the Qwen 2.5 7B base model, and PPO refers to a PPO-trained variant of the same 7B model. For both CD and Q♯, a Qwen 2.5 1.5B model was used to estimate value functions and guide the 7B policy during inference. We apologize for the confusion the reviewer might have about model sizes in our evaluation.
> > >
> > > In terms of training setup: all methods (CD, Q♯, PPO) were trained on the MATH training set, with checkpoint selection performed on a held-out validation portion of MATH. No further fine-tuning was done on AIME-24. Evaluations were conducted directly on AIME to test generalization.
> > >
> > > To reduce noise in our evaluation, for each AIME problem, we generate 8 responses from the model. We report: *pass@1* as the empirical mean success rate across problems, estimated via bootstrapping; *maj@8* as the accuracy from majority voting over the 8 responses; *KL-divergence* as the average KL divergence from the reference policy across generated responses.
> > >
> > > The results below show that PPO underperforms CD and Q♯ on both metrics and exhibits significantly higher KL divergence, indicating reduced alignment with the reference policy. This supports our earlier hypothesis (Q1) that PPO, while strong on some tasks, may generalize less effectively.
> > >
> > > |AIME-24||Qwen2.5 7B|||
> > > |-|:-:|:-:|:-:|:-:|
> > > |Methods|$\pi^{\text{ref}}$|CD|$Q\sharp$|PPO|
> > > |pass@1 ↑|9.3|13.5|**14.1**|10.9|
> > > |maj1@8 ↑|16.7|16.7|**20.0**|13.3|
> > > |KL-Divergence ↓|-|9.95|**9.23**|91.89|
> > >
> > > Thank you again for your continued feedback on our work. Please don’t hesitate to let us know if any questions remain or if further clarification would be helpful.
> > >
> > > **Citations**
> > >
> > > [1] Casper, Stephen, et al. "Open Problems and Fundamental Limitations of Reinforcement Learning from Human Feedback." Transactions on Machine Learning Research.
> > >
> > > [2] Kirk, Robert, et al. "Understanding the Effects of RLHF on LLM Generalisation and Diversity." The Twelfth International Conference on Learning Representations.

---

> > > > ### Comment · Reviewer_H6aA · 2025-08-05
> > > >
> > > > Thanks for the authors' clarification. I will increase my score to 4.

---

> > > > > ### Author Response · Authors · 2025-08-05
> > > > > **Response to Reviewer H6aA**
> > > > >
> > > > > Dear Reviewer H6aA,
> > > > >
> > > > > We sincerely appreciate you increasing your recommendation. We will carefully incorporate your suggestions and the additional results into the revised version of the paper.

---

### Official Review · Reviewer_3hUn · 2025-06-29

**Clarity:** 3
**Significance:** 3
**Originality:** 2
**Rating:** 5
**Confidence:** 3

**Summary:**

The authors introduce $Q\sharp$ a novel RL algorithm for LLM post-training. $Q\sharp$  works in deterministic MDPs and makes use of value learning. Unlike other value learning approaches, $Q\sharp$  doesn't use a TD objective. The authors show bounds of the algorithm as well as empirical results. Indeed $Q\sharp$ beats the Controlled Decoding algorithm (CD) with relative consistency (although the improvements are very marginal.)

**Questions:**

* Is the performance difference between CD and $Q\sharp$ significant?
* How does $Q\sharp$ compare to other algorithms like GRPO on math reasoning tasks?
* The authors note that their algorithm works for scenarios with deterministic transition dynamics. Can the authors highlight an environment with stochastic dynamics in the limitations to make it more intuitive for the reader what the limitations of the algorithm are?

**Ethical Concerns:**

["NO or VERY MINOR ethics concerns only"]

**Final Justification:**

Authors included extra evaluations and implemented an additional baseline, so I've increased my score to 5.

**Limitations:**

Yes

**Quality:**

2

**Strengths And Weaknesses:**

Strengths:
* The algorithm is novel and improves upon the baseline presented (CD).
* Value estimation in an LLM-RL setting is an important and challenging topic.
* PAC bounds are provided.
* Algorithm tested with both Llama and Qwen family of models.

Weaknesses:
* There are some concerns around the robustness of the evaluations on GSM8k and MATH500. There are no mentions of the number of seeds used to train the models (I'm assuming just a single seed), and it's therefore impossible to know whether the difference in score is significant, or how robust the learning algorithms are.
* It's also unclear how this method compares to other household algorithm, both in terms of training speed and final converged performance. Here, adding another baseline line like GRPO would add a lot of scientific value to the paper.

Given the above strengths and weaknesses I can only recommend borderline accept, but I'm happy to increase my score if the authors show results with more seeds and another baseline (ideally something like GRPO).

---

> ### Author Rebuttal · Authors · 2025-07-30
>
> Dear Reviewer 3hUn,
>
> Thank you for your thoughtful feedback and we respond to individual points below.
>
> > There are some concerns around the robustness of the evaluations on GSM8k and MATH500. There are no mentions of the number of seeds used to train the models (I'm assuming just a single seed), and it's therefore impossible to know whether the difference in score is significant, or how robust the learning algorithms are.
>
> > Is the performance difference between CD and Q♯ significant?
>
> **Authors' reply:** Thank you for raising this important point. For a direct comparison between CD and Q♯, we used the same trained model and varied only the value calculation used at inference. As a result, the primary source of randomness in evaluation comes from sampling variability during generation. To quantify this, we computed standard deviations via empirical bootstrapping over the GSM8K test set. For pass@1 and maj1@8, the standard deviations are approximately 0.7% and 0.3% respectively, suggesting that the performance gains of Q♯ over CD are still likely to be meaningful.
>
> To further assess robustness and generality, we assess CD and Q♯ performance on two additional benchmarks: AIME-24, a more challenging math reasoning dataset, and QuALITY [1], a multiple-choice QA benchmark with context passages from Project Gutenberg. On AIME-24, we used Q♯ (Qwen 2.5 1.5B) trained on MATH to guide Qwen 2.5 7B. For QuALITY, we trained with two settings: Qwen 2.5 1B guiding Qwen 2.5 7B and Llama 3.2 1B guiding Llama 3.1 8B.
>
> |Dataset||AIME-24||
> |-|:-:|:-:|:-:|
> |Methods|$\pi^{\text{ref}}$|CD|$Q\sharp$|
> |pass@1 ↑|9.3|13.5|**14.1**|
> |maj1@8 ↑|16.7|16.7|**20.0**|
> |KL-Divergence ↓|-|9.95|**9.23**|
>
> |$\pi^{\text{ref}}$||Qwen 2.5 7B|||Llama 3.1 8B||
> |-|:-:|:-:|:-:|:-:|:-:|:-:|
> |Methods|$\pi^{\text{ref}}$|CD|$Q\sharp$|$\pi^{\text{ref}}$|CD|$Q\sharp$|
> |pass@1 ↑|64.5|64.2|**68.1**|73.5|75.1|**75.9**|
> |maj1@8 ↑|72.0|66.3|**73.3**|79.3|79.3|**81.1**|
> |KL-Divergence ↓|-|12.32|**7.90**|-|9.23|**8.88**|
>
> These results across multiple datasets and model families provide evidence that Q♯ is generally robust and can consistently improve over both the reference policy and CD, while maintaining lower KL divergence.
>
> > It's also unclear how this method compares to other household algorithm, both in terms of training speed and final converged performance. Here, adding another baseline line like GRPO would add a lot of scientific value to the paper.
>
> > How does Q♯ compare to other algorithms like GRPO on math reasoning tasks?
>
> **Authors' reply:** Thank you for the helpful suggestion and we conduct additional experiments on the MATH dataset using PPO as a household algorithm to compare performance and training speed.
>
> In these experiments, Q♯ is instantiated with a Qwen 2.5 1.5B model and used to guide both: (1) the original Qwen 2.5 7B reference model, and (2) the same model after PPO post-training. As shown in the table below, Q♯ consistently improves performance in both settings. Notably, when applied to the PPO-trained policy, Q♯ further increases accuracies and reduces the KL divergence — indicating improved alignment with $\pi^{\text{ref}}$ and better performance. The results suggest that Q♯ can also be complementary to policy-based approaches.
>
> We additionally note a qualitative distinction: PPO improves pass@1 but slightly reduces maj1@8, indicating that its generations tend to be lower entropy and less diverse. Q♯, in contrast, improves both metrics.
>
> |Dataset|MATH||||
> |-|:-:|:-:|:-:|:-:|
> |Methods|$\pi^{\text{ref}}$|$\pi^{\text{ref}}$ + $Q\sharp$|PPO|PPO + $Q\sharp$|
> |pass@1 ↑|58.6|61.9|68.4|**71.1**|
> |maj1@8 ↑|72.8|**74.8**|72.4|73.4|
> |KL-Divergence ↓|-|**6.46**|69.52|60.53|
>
> In terms of training speed, Q♯ is significantly lighter. PPO requires approximately 20 hours on 4×H100 GPUs, whereas Q♯ training completes in roughly 5 hours on a single H100 GPU, thanks to its supervised learning objective and the use of a much smaller model. These findings suggest that Q♯ can effectively enhance performance, demonstrating its practical advantage as a complementary lightweight module.
>
> > The authors note that their algorithm works for scenarios with deterministic transition dynamics. Can the authors highlight an environment with stochastic dynamics in the limitations to make it more intuitive for the reader what the limitations of the algorithm are?
>
> **Authors' reply:** One example environment with stochastic dynamics in the context of LLMs is multi-agent interaction, such as game playing with another external player. While the LLM can generate its own actions, the next state depends on the (potentially unpredictable) behavior of the other player, making the overall transition dynamics stochastic. We will include this example in the Limitations section of the final version to make the scope of our assumptions more concrete and intuitive.
>
> We will update the final version of the paper with the above discussion and evaluation results. Please let us know if you have any further questions!
>
> **Citations**
>
> [1] Pang, Richard Yuanzhe, Alicia Parrish, Nitish Joshi, Nikita Nangia, Jason Phang, Angelica Chen, Vishakh Padmakumar et al. "QuALITY: Question Answering with Long Input Texts, Yes!." In Proceedings of the 2022 Conference of the North American Chapter of the Association for Computational Linguistics: Human Language Technologies, pp. 5336-5358. 2022.

---

> > ### Comment · Reviewer_3hUn · 2025-08-04
> >
> > Thanks for answering my questions and extending the evaluation, I've increased my score to 5.

---

> > > ### Author Response · Authors · 2025-08-04
> > > **Response to Reviewer 3hUn**
> > >
> > > Dear Reviewer 3hUn,
> > >
> > > Thank you for taking the time reading our rebuttal and increasing your recommendation. We will make sure to incorporate your suggestions and the additional results into the revised version of the paper.

---

### Official Review · Reviewer_bUKk · 2025-07-02

**Clarity:** 3
**Significance:** 3
**Originality:** 3
**Rating:** 5
**Confidence:** 3

**Summary:**

This paper proposes Q♯, a new value-based algorithm for KL-regularized RL, tailored for post-training LLMs. Unlike popular policy-based methods (like PPO, DPO) that often fail to correct shortcuts learned during pre-training, Q♯ learns the optimal KL-regularized Q-function using distributional RL, then uses it to guide the reference policy. The approach treats LLM post-training as a deterministic MDP, fitting well with the autoregressive nature of language generation. Theoretically, it reduces KL-regularized RL to no-regret online learning, providing variance-dependent PAC bounds that guarantee convergence. Empirically, on tasks like star-graph planning, GSM8K, and MATH, Q♯ outperforms existing baselines, achieving higher reasoning accuracy with lower KL divergence from the reference policy.

**Questions:**

I think this paper has a strong theoretical foundation, but it would be even stronger with more empirical support. For example:
- How does the empirical performance of Q♯ compare to policy-based methods (like PPO and DPO) on GSM8K and MATH?
- How does Q♯ compare empirically to VAS?

**Ethical Concerns:**

["NO or VERY MINOR ethics concerns only"]

**Final Justification:**

After reading the rebuttal and carefully considering the authors' responses, I have decided to raise my score to a 5.

**Limitations:**

yes

**Paper Formatting Concerns:**

Not applicable.

**Quality:**

4

**Strengths And Weaknesses:**

### **Strengths**

- While previous value-based approaches (like CD, VAS) guided policies using unregularized Q-functions (which does not solve the KL-regularized objective), Q♯ explicitly uses the optimal regularized Q⋆,η and provides rigorous convergence guarantees.
- Avoids unstable TD bootstrapping by directly learning the cumulative reward distribution, framing it as supervised learning. This leads to better stability and faster convergence, especially when the reference policy has low variance.
- On the star-graph task, Q♯ corrects the "Clever Hans" shortcut that policy-based methods like REINFORCE or DPO fail to fix.
- On GSM8K and MATH datasets, Q♯ consistently improves pass@1 and majority voting accuracy over CD and the original reference model, while also incurring lower KL divergence.
- Unlike CD, whose performance is highly sensitive to η and can suffer large KL spikes, Q♯ is empirically much more stable, reducing the burden of hyperparameter tuning.
- The learned Q♯ can also evaluate complete generations as a reward model, improving Best-of-N sampling and providing generalizable scoring across different generations.

&nbsp;

### **Weakness**
What I understood as the claimed advantages of the proposed method over policy-based methods:
(1) It does not require updating all the large model parameters, only needs to learn the Q function.
(2) It can correct biases and shortcuts learned during pre-training.

What I’m most curious about is:
Why can’t policy-based methods fix such shortcuts, whereas Q♯ can?
Section 3.1’s star-graph experiment is interesting, but it’s a simple toy environment, and the paper essentially says “it worked when we tried it.” It does not clearly explain why only Q♯ is able to correct these shortcuts. I’d like to understand the reason behind this more deeply.

Also, what would be an example of such a shortcut (or hacky solution) in math? Is there an example from GSM8K or MATH where Q♯ uniquely fixes a mathematical shortcut that the policy-based methods fail to fix?

Looking at Appendix G’s qualitative examples, it seems that the benefits of Q♯ come more from general advantages of LLM post-training — such as fixing calculation errors, misunderstanding the problem-solving process, or skipping essential logical steps — rather than specifically correcting “shortcuts.” So it’s still unclear to me in what empirical ways Q♯ is actually better than policy-based methods.

Along similar lines, the star-graph planning task, GSM8K, and MATH actually seem to be quite different in nature. The star-graph task appears to require more explicit planning, while GSM8K and MATH seem to focus on reasoning. Because of this, showing a toy experiment on star-graph planning and then moving on to GSM8K and MATH feels a bit disconnected.

---

> ### Author Rebuttal · Authors · 2025-07-30
>
> Dear Reviewer bUKk,
>
> Thank you so much for your detailed review and suggestions for us to improve the paper. We respond to individual points below.
>
> > What I understood as the claimed advantages of the proposed method over policy-based methods: (1) It does not require updating all the large model parameters, only needs to learn the Q function. (2) It can correct biases and shortcuts learned during pre-training.
>
> > How does the empirical performance of Q♯ compare to policy-based methods (like PPO and DPO) on GSM8K and MATH?
>
> **Authors' reply:** Thank you for the helpful question and the understanding is correct. We also add that value-based methods like Q♯ can serve as a lightweight complement to policy-based approaches. Specifically, Q♯ can guide both the base reference policy and policies trained via reinforcement learning such as PPO.
>
> To empirically assess this, we present results on the MATH dataset where Q♯ is instantiated with a Qwen 2.5 1.5B model and used to guide: (1) the base Qwen 2.5 7B reference model and (2) a PPO-trained version of the same model. As shown in the table below, Q♯ consistently improves both pass@1 and maj1@8 for each policy. In particular, when applied to the PPO-trained policy, Q♯ reduces the KL divergence from the reference policy while further boosting accuracy. We also note a qualitative distinction: PPO improves pass@1 but slightly reduces maj1@8, indicating that its generations tend to be lower entropy and less diverse. Q♯, in contrast, improves both metrics while maintaining closer alignment with the reference.
>
> |Dataset|MATH||||
> |-|:-:|:-:|:-:|:-:|
> |Methods|$\pi^{\text{ref}}$|$\pi^{\text{ref}}$ + $Q\sharp$|PPO|PPO + $Q\sharp$|
> |pass@1 ↑|58.6|61.9|68.4|**71.1**|
> |maj1@8 ↑|72.8|**74.8**|72.4|73.4|
> |KL-Divergence ↓|-|**6.46**|69.52|60.53|
>
> In terms of efficiency, Q♯ is significantly lighter to train. PPO requires approximately 20 hours on 4×H100 GPUs, whereas Q♯ training completes in roughly 5 hours on a single H100 GPU, thanks to its supervised learning objective and the use of a much smaller model. These findings suggest that Q♯ can effectively enhance performance while maintaining closer alignment with the reference policy, demonstrating its practical advantage as a complementary lightweight module.
>
> **Why Q♯ can fix shortcuts while REINFORCE and RPO cannot**
>
> **Authors' reply:** Thank you for the insightful question. The shortcut behavior, also known as the Clever Hans Trick [1], in the star-graph task arises directly from the auto-regressive **next-token prediction** objective. Specifically, the model minimizes loss by memorizing the first token seen during training and following the corresponding edge, achieving low training error but generalizing poorly at test time when the **initial** token is not provided. This leads to a brittle, shortcut-based policy.
>
> Policy-based methods such as REINFORCE and RPO attempt to correct this by upweighting high-reward trajectories. However, because their loss is still based on the product of next-token prediction probabilities, the same as in pretraining, they are vulnerable to the same shortcut and require **exponentially many samples** via gradient descent on the policy to correct it once it is learned (Theorem 1 of [2]).
>
> In contrast, Q♯ does not depend on myopic next-token supervision. Instead, it learns to predict the **cumulative reward-to-go** from each (prefix, token) pair under the reference policy, and uses this to guide generation toward optimal completions. This token-level value modeling allows Q♯ to predict future outcome and assign higher value to early tokens that lead to long-term reward. In other words, Q♯’s loss function is directly trained to perform *planning*, making it robust to the Clever Hans Trick [1] that undermines next-token–based methods. As shown in Appendix Figure 5, both Q♯ and CD are able to solve the star-graph task near-perfectly, while policy-based methods perform at random-guess level. We will incorporate this discussion into the revised version.
>
> > Also, what would be an example of such a shortcut (or hacky solution) in math? Is there an example from GSM8K or MATH where Q♯ uniquely fixes a mathematical shortcut that the policy-based methods fail to fix?
>
> **Authors' reply:** In our experiments on GSM8K and MATH, we did not observe specific, easily isolatable shortcuts of the same form as those seen in the star-graph setting. This may be due to the more flexible and open-ended nature of mathematical reasoning.
>
> To provide intuition, we describe a contrived example inspired by integration by parts. Consider the problem $\int x e^x , dx$, which requires selecting the appropriate decomposition into $u$ and $dv$. One path (e.g., choosing $u = x$, $dv = e^x dx$) leads to a clean and solvable integral. The other path (e.g., $u = e^x$, $dv = x dx$) results in a recursive or more complex expression. A model trained purely via next-token prediction loss may mimic observed solutions without internalizing the decision-making criteria, potentially choosing either path arbitrarily. In contrast, a value-based approach like Q♯ can learn to distinguish between such paths based on expected reward, consistently selecting the option that leads to successful solutions.
>
> **Benefits of Q♯ from qualitative examples**
>
> **Authors' reply:** Thank you for the detailed observation. Our goal with the qualitative examples are mainly to show that Q♯ can correct the reference policy’s mistakes while maintaining low KL divergence. As shown, Q♯ often starts with a similar prefix as $\pi^{\text{ref}}$, but when a divergence occurs, it typically avoids errors that $\pi^{\text{ref}}$ commits—such as arithmetic mistakes or flawed logical steps. This behavior reflects Q♯’s ability to assign higher value to correct tokens, thereby steering generation more effectively at critical decision points. We will clarify the description of qualitative examples more clearly in the final version.
>
> > Along similar lines, the star-graph planning task, GSM8K, and MATH actually seem to be quite different in nature. The star-graph task appears to require more explicit planning, while GSM8K and MATH seem to focus on reasoning. Because of this, showing a toy experiment on star-graph planning and then moving on to GSM8K and MATH feels a bit disconnected.
>
> **Authors' reply:** We agree with the reviewer that the star-graph task differs in nature from math reasoning benchmarks like GSM8K and MATH. Our intention in including the star-graph experiment is to use it as a controlled and interpretable environment to highlight a core limitation of next-token prediction loss. The star-graph setting clearly illustrates how value-based methods, such as Q♯, can overcome this limitation by considering long-term outcomes.
>
> We then turn to GSM8K and MATH to evaluate Q♯ in more realistic and complex reasoning tasks. While these domains do not exhibit the same explicit planning structure as the star-graph, they still involve critical decision points where early mistakes can propagate—analogous in spirit to the planning failures observed in star-graph. Our empirical results across multiple model sizes demonstrate that Q♯ consistently outperforms CD in these benchmarks, reinforcing the broader applicability and effectiveness of our value-based approach.
>
> > How does Q♯ compare empirically to VAS?
>
> **Authors' reply:** Thank you for the question. CD and VAS are concurrent works that adopt the same core approach: both use the unregularized Q function $Q^{\pi^{\text{ref}}}$—i.e., the expected reward-to-go under $\pi^{\text{ref}}$ without KL regularization—to reweight the reference policy. As we show in Section 4.1, this choice leads to fundamental theoretical limitations: neither CD nor VAS guarantees convergence to the optimal KL-regularized policy, and both can suffer from suboptimal reward and excessive KL divergence.
>
> In terms of implementation, CD and VAS differ only in minor aspects such as sampling strategy (e.g., top-k filtering), but these do not affect the underlying objective. As a result, we treat them as representative of the same class of methods, and use CD as a canonical baseline for empirical comparison. We will clarify this connection more explicitly in the final version of the paper.
>
> **Citations**
>
> [1] Bachmann, Gregor, and Vaishnavh Nagarajan. "The pitfalls of next-token prediction." arXiv preprint arXiv:2403.06963 (2024).
>
> [2] Hu, Edward S., Kwangjun Ahn, Qinghua Liu, Haoran Xu, Manan Tomar, Ada Langford, Dinesh Jayaraman, Alex Lamb, and John Langford. "The belief state transformer." arXiv preprint arXiv:2410.23506 (2024).

---

> > ### Comment · Reviewer_bUKk · 2025-08-06
> >
> > Thank you to the authors for the detailed explanation. Most of my concerns have been addressed, and I have raised my assessment to a 5.
> >
> > As an additional comment, the reason I asked about the strengths of Q♯ in the qualitative examples in Appendix G is that, based on the appendix, the cases Q♯ solves appear to involve avoiding arithmetic mistakes or logical errors, as the authors noted. However, improvements from fixing calculation errors or skipping essential logical steps could potentially also be achieved by methods like PPO. While I understand the authors’ point that Q♯ is more computationally efficient compared to PPO, given the paper’s claim that Q♯ can address issues like shortcuts that arise in conventional training, it would be helpful if the final version included more qualitative analysis that clearly demonstrates such benefits in a real math example.

---

> > > ### Author Response · Authors · 2025-08-06
> > > **Response to Reviewer bUKk**
> > >
> > > Dear Reviewer bUKk,
> > >
> > > Thank you for your valuable time and suggestions. We are glad to hear that our rebuttal addressed most of your concerns and appreciate your updated assessment. We will clarify the descriptions of existing qualitative examples and include additional analysis comparing Q♯ with PPO generations in the final version of the paper.

---

### Official Review · Reviewer_2Dby · 2025-07-05

**Clarity:** 3
**Significance:** 3
**Originality:** 3
**Rating:** 4
**Confidence:** 2

**Summary:**

This paper proposes Q#, a novel distributional reinforcement learning algorithm designed for post-training LLMs. It addresses shortcomings of existing methods like PPO and DPO, which fail to fix shortcuts inherited from pre-training. By introducing a value-based approach, Q# uses a KL-regularized Q-function to guide the model without modifying its weights. The method is shown to outperform previous baselines in math reasoning tasks while maintaining a lower KL divergence from the reference policy.

**Questions:**

1. What would the performance look like in real-world LLM tasks beyond math reasoning (e.g., commonsense reasoning)?
2. How sensitive is Q# to hyperparameter choices and is there a trade-off between performance and stability in different configurations?

**Ethical Concerns:**

["NO or VERY MINOR ethics concerns only"]

**Final Justification:**

After reading the author's rebuttal, I have decided to keep my score unchanged.

**Limitations:**

Yes

**Quality:**

3

**Strengths And Weaknesses:**

Strength:
1. The paper provides provable guarantees for the convergence of the Q# algorithm, a significant theoretical advancement in KL-regularized RL, which ensures that the method avoids pitfalls of previous approaches.
2. Q# consistently outperforms existing methods like CD and VAS in benchmarks, showing its ability to fix biases and improve performance in math reasoning tasks.
3. The algorithm shows that smaller models can effectively guide large models, which could lead to more resource-efficient post-training approaches.

Weakness:
1. The experiments focus heavily on math reasoning tasks and a star-graph environment, leaving other potential applications of Q# unexplored.
2. While Q# works with models as small as 1B parameters, the authors do not explore the computational overhead involved when scaling up to very large LLMs (e.g., 70B parameters).

---

> ### Author Rebuttal · Authors · 2025-07-30
>
> Dear Reviewer 2Dby,
>
> Thank you for your constructive feedback and we respond to individual points below.
>
> **Other potential applications of Q♯ beyond math reasoning**
>
> **Authors' reply:** Thank you for the valuable comment! Following your suggestion, we conducted experiments on QuALITY [1], a multiple-choice QA benchmark with context passages in English sourced from Project Gutenberg. We experimented with two settings: Qwen 2.5 1B guiding Qwen 2.5 7B and Llama 3.2 1B guiding Llama 3.1 8B.
>
> |$\pi^{\text{ref}}$||Qwen 2.5 7B|||Llama 3.1 8B||
> |-|:-:|:-:|:-:|:-:|:-:|:-:|
> |Methods|$\pi^{\text{ref}}$|CD|$Q\sharp$|$\pi^{\text{ref}}$|CD|$Q\sharp$|
> |pass@1 ↑|64.5|64.2|**68.1**|73.5|75.1|**75.9**|
> |maj1@8 ↑|72.0|66.3|**73.3**|79.3|79.3|**81.1**|
> |KL-Divergence ↓|-|12.32|**7.90**|-|9.23|**8.88**|
>
> The results suggest that Q♯ can also consistently improve performance on QA tasks for models such as Qwen 2.5 and Llama 3.1 with smaller KL divergence than CD baseline.
>
> > While Q♯ works with models as small as 1B parameters, the authors do not explore the computational overhead involved when scaling up to very large LLMs (e.g., 70B parameters).
>
> **Authors' reply:** Thank you for the note. We would like to clarify that our approach is designed to efficiently guide large LLMs (such as Llama 70B) using much smaller Q♯ models (1B or 3B parameters), rather than scaling Q♯ itself to the same size as reference policy. As shown in Table 2, even a 1B Q♯ model improves the performance of a 70B reference model on MATH, both in terms of pass@1 and majority voting accuracy.
>
> We also analyze runtime implications in Appendix H. The overhead incurred by Q♯ is approximately proportional to the size ratio between the Q♯ model and the reference model. This overhead becomes relatively smaller as the reference model scales up. For instance, when guiding Llama 8B with a 1B Q♯ model using our Huggingface Transformers implementation, the additional runtime is around 20%. This cost remains modest even in larger-scale settings and is justified by the observed performance gains.
>
> **Sensitivity of Q♯ to hyperparameter choices**
>
> **Authors' reply:** Thank you for raising this important point. Q♯ demonstrates strong robustness to hyperparameter choices, especially when compared to the CD baseline. As shown in Figure 3, the performance of CD is highly sensitive to the choice of $\eta$: as $\eta$ decreases, CD’s pass@1 accuracy drops sharply while its KL divergence increases significantly. In contrast, Q♯ maintains stable performance across a range of $\eta$ values and consistently achieves a more favorable trade-off between accuracy and KL divergence. This indicates that Q♯ not only provides improved performance but also greater stability under different configurations, reducing the need for extensive hyperparameter tuning.
>
> **Citations**
>
> [1] Pang, Richard Yuanzhe, Alicia Parrish, Nitish Joshi, Nikita Nangia, Jason Phang, Angelica Chen, Vishakh Padmakumar et al. "QuALITY: Question Answering with Long Input Texts, Yes!." In Proceedings of the 2022 Conference of the North American Chapter of the Association for Computational Linguistics: Human Language Technologies, pp. 5336-5358. 2022.

---

> ### Comment · Reviewer_2Dby · 2025-08-08
>
> After reading the author's rebuttal, I have decided to keep my score unchanged.

---

> > ### Author Response · Authors · 2025-08-08
> > **Response to Reviewer 2Dby**
> >
> > Dear Reviewer 2Dby,
> >
> > Thank you for taking the time to read our rebuttal and for maintaining your positive assessment of our work. We will incorporate the rebuttal discussion and results in the revised version of the paper.

---

### Decision · Program_Chairs · 2025-09-17

**Decision:**

Accept (poster)

**Comment:**

This paper introduces Q#, a novel and theoretically principled value-based algorithm for the KL-regularized RL post-training of LLMs. The core contribution is leveraging distributional RL to directly learn the optimal KL-regularized Q-function, which provides a more sound objective than prior value-based methods like CD and VAS. The reviewers all agreed that the paper has a strong theoretical grounding , where the authors provided PAC bounds under a mild realizability assumption. Some concerned were raised regarding the empirical evaluation, requesting comparisons to stronger policy-based baselines like PPO and broader testing on more diverse and challenging datasets. The authors addressed these points in their rebuttal, with new experiments on the AIME and QuALITY benchmarks and a direct comparison against PPO. Therefore, I recommend accepting this paper, contingent on the authors incorporating the valuable new experiments, analysis, and clarifications from their rebuttal into the final camera-ready version.